# Three Copies of *zbed1* Specific in Chromosome W Are Essential for Female-Biased Sexual Size Dimorphism in *Cynoglossus semilaevis*

**DOI:** 10.3390/biology13030141

**Published:** 2024-02-23

**Authors:** Yuqi Sun, Xihong Li, Jiaqi Mai, Wenteng Xu, Jiacheng Wang, Qi Zhang, Na Wang

**Affiliations:** 1Jiangsu Key Laboratory of Marine Bioresources and Environment, Jiangsu Ocean University, Lianyungang 222000, China; sunyuqi2099926838@163.com; 2National Key Laboratory of Mariculture Biobreeding and Sustainable Goods, Yellow Sea Fisheries Research Institute, Chinese Academy of Fishery Sciences, Qingdao 266071, China; lixh@ysfri.ac.cn (X.L.); 15999140520@163.com (J.M.); xuwt@ysfri.ac.cn (W.X.); 18653462716@163.com (J.W.); zqmtdzjbz@foxmail.com (Q.Z.); 3College of Fisheries and Life Science, Dalian Ocean University, Dalian 116023, China; 4College of Fisheries and Life Science, Shanghai Ocean University, Shanghai 201306, China; 5Fisheries College, Zhejiang Ocean University, Zhoushan 316022, China

**Keywords:** *Cynoglossus semilaevis*, sexual size dimorphism, *zbed1*

## Abstract

**Simple Summary:**

The sex chromosome, especially specific in one sex, is considered to determine the sexual size dimorphism (SSD), a dimorphic sexual difference in the body size. For *Cynoglossus semilaevis*, a flatfish unique in China, the important role of female-specific chromosome W in its female-biased SSD was previously implied. Furthermore, a W chromosome-specific gene *zbed1* was identified to potentially regulate female-biased SSD in *C. semilaevis*. However, the chromosome’s location, family members, and detailed network information are still unknown. At present, the genome-wide identification of ZBED family members and dPCR experiment both confirm that three copies of the *zbed1* gene are located in chromosome W, with no *zbed1* gene in chromosome Z. Phylogenetic analysis for ZBED family revealed an existence of ZBED9 in the fish. Nine members were uncovered from *C. semilaevis*, clustering into three kinds, ZBED1, ZBED4 and ZBEDX, which is less than the eleven kinds of ZBED members in mammals. For the chromosome-W-specific *zbed1,* qPCR verified its predominant expression in the female brain and pituitary tissues. The dual luciferase activity test showed that transcription factor *c/ebpα* could significantly enhance the transcriptional activity of *zbed1* promoter, which is opposite to its effect on the male determinant factor *dmrt1*. In addition, after *zbed1* interfered in the brain cells, *piwil1*, *esr2* and *wnt7b* were up-regulated, while cell-cycle-related genes (*tbp*, *cdk2*, *cdk4*, *cdk6*, *ccng1* and *ccndx*) were down-regulated. It is suggested that the cell proliferation function of zbed1 may be realized by regulating *esr2*, *piwil1*, cell cycle and the Wnt pathway.

**Abstract:**

The sex chromosome, especially specific in one sex, generally determines sexual size dimorphism (SSD), a phenomenon with dimorphic sexual difference in the body size. For *Cynoglossus semilaevis*, a flatfish in China, although the importance of chromosome W and its specific gene *zbed1* in female-biased SSD have been suggested, its family members and regulation information are still unknown. At present, three *zbed1* copies gene were identified on chromosome W, with no gametologs. Phylogenetic analysis for the ZBED family revealed an existence of ZBED9 in the fish. Nine members were uncovered from *C. semilaevis*, clustering into three kinds, ZBED1, ZBED4 and ZBEDX, which is less than the eleven kinds of ZBED members in mammals. The predominant expression of *zbed1* in the female brain and pituitary tissues was further verified by qPCR. Transcription factor *c/ebpα* could significantly enhance the transcriptional activity of *zbed1* promoter, which is opposite to its effect on the male determinant factor-*dmrt1*. When *zbed1* was interfered with, *piwil1*, *esr2* and *wnt7b* were up-regulated, while cell-cycle-related genes, including *cdk4* and *ccng1,* were down-regulated. Thus, *zbed1* is involved in cell proliferation by regulating *esr2*, *piwil1*, cell cycle and the Wnt pathway. Further research on their interactions would be helpful to understand fish SSD.

## 1. Introduction

The phenomenon of body size varying greatly between females and males is known as sexual size dimorphism (SSD), which exists widely across multiple species, including arthropods, fish, reptiles, birds, and mammals [1,2,3]. The formation of SSD is usually related to the allometric growth and natural selection in different sexes [4]. Sex steroids and the somatotrophic axis (mainly GH-IGF1 system) have been reported to influence the sexual allometric growth [5,6,7].

Since 1984, Rice and others have put forward the theory of “sex chromosome dominating gender abnormality” [8]. Increasing evidence has indicated that the sex chromosome may be a dominant factor for SSD. For instance, mice with two X chromosomes have characteristics such as heavier body weight than mice with only one X chromosome [9]. In *Portuguese Water Dogs*, the interaction of CHM marker on the chromosome X and *igf1* gene on the chromosome 15 could result in male-biased SSD [10]. Studies have also found that the singing control area of the male brain of *zebra finch* is usually 5–6 times larger than that of the female [11,12], which is mainly controlled by sex chromosomes [13]. The sexual dimorphic singing behavior of another bird, the white-waisted wingbird, was regulated by the Z-chromosome-linked gene *erbin* through its influence on brain cell proliferation and neuronal differentiation [14]. In *drosphila*, except for the classical insulin/IGF pathway (IIS) [7,15], the sex-determining gene *sxl* in chromosome X has also been proven to regulate female-biased SSD by cell-autonomy and non-cell-autonomy mechanisms [16,17]. Yet, in the seed beetle, a Y-linked *TOR* provides a male-specific opportunity to affect male body size and thereby sexual size dimorphism (SSD) [18].On the other hand, there is no significant correlation between sex chromosome types of radial fin fish and fin decoration characteristics of male fish [19,20]. In view of this, it is worth investigating whether a sex chromosome plays a role in the female-biased SSD of the Chinese female heterogamete flatfish, *Cynoglossus semilaevis* (ZW♀/ZZ♂) [21,22].

Our previous transcriptomic analysis of female and male *C. semilaevis* has screened 204 genes on chromosome W [23], implying the important role of chromosome W in its female-biased SSD. Further GWAS has identified the chromosome-W-linked transcription factor zinc finger BED domain-containing protein 1 (*zbed1*) from *C. semilaevis* [24]. The *Zbed1* gene, a homologue of *DREF* in Drosophila, was firstly identified by Ohshima in 2003 as a key transcription factor for cell proliferation and can participate in DNA replication and cell proliferation by regulating histone H1 ribosomal protein RP [25,26].

In mammals, eleven ZBED family members have been identified and derived from “molecularly domesticated” hAT DNA transposons, which are divided into two sub-families (ZBED5 and ZBED7–9 belong to the Buster sub-family, while the others were assigned to Ac sub-family) [27]. Besides *zbed1*, other members, including *zbed2*, *zbed3, zbed4*, and *zbed6*, are also involved in regulating brain development, growth, insulin metabolism, interferon response pathway, and so on [28,29,30,31,32]. For example, the *zbed3* protein assumes a pivotal function in the modulation of the Wnt/β-catenin pathway in mammalian organisms [31], *zbed4* is closely related to retinal morphogenesis and isopathways [29], and *zbed6* can regulate cell proliferation, differentiation and growth by inhibiting *igf2* and other genes [30].

However, only three members of the ZBED family, ZBED1, ZBED4 and ZBEDX, have been identified in most fish [33], which seems to be contradictory to fish-specific genome duplication [34]. Until now, only two members of the ZBED family, ZBED1 and ZBED4, have been found in *C. semilaevis*. Whether there are other members is not clear.

Therefore, the present study aims to firstly identify *C. semilaevis* ZBED family members from a genome-wide perspective, which is also helpful to confirm whether *zbed1* has its homologue gene in chromosome Z and whether any other members exist in this fish. The spatiotemporal expression pattern and transcriptional regulation of *zbed1* are subsequently examined. Finally, the knockdown effect of *zbed1* is studied to reveal its molecular pathways involved in sexual size dimorphism.

## 2. Materials and Methods

### 2.1. Ethics Statement

MS-222 was used to anesthetize fish individuals prior to the experiments in the present study. The Animal Care and Use Committee of the Yellow Sea Fisheries Research Institute, Chinese Academy of Fishery Sciences approved the sampling and treatment of *C. semilaevis* in the present study.

### 2.2. Cell Culture and Transfection

HEK 293T cells derived from human embryonic kidneys and *C. semilaevis* brain cells obtained from female individuals were utilized in this study. HEK 293T cells were cultured in DMEM medium supplemented with 10% fetal bovine serum (FBS) (Bovogen, Melbourne, Australia), and maintained at 37 °C with 5% CO_2_. Additionally, 1% antibiotics were added to the culture. *C. semilaevis* cells were cultured in L-15 medium supplemented with 10% FBS, 5 ng/mL bFGF (Beyotime, Shanghai, China), 5 ng/mL LIF (Beyotime, Shanghai, China), and 1% antibiotics at 24 °C [35]. The day before transfection, the cells were incubated in 24-well or 12-well plates with a density ranging from 60% to 80%. Lipo8000^TM^ Transfection Reagent (Beyotime, Shanghai, China) and riboFECT^TM^ CP Transfection Kit (Ribobio, Guangzhou, China) were used for the transfection of plasmids and small interfering RNAs (siRNAs), respectively [36]. 

### 2.3. Experimental Samples

The *C. semilaevis* samples were obtained from Haiyang Yellow Sea Fisheries Limited Company in Shandong, and the genetic sex was determined combining visual observation and a molecular identification method with the previously reported primers Cs-SEX-F and Cs-SEX-R (Table 1) [37]. After dissection, the gonad, kidney, intestine, brain, pituitary, gill, spleen, muscle, skin, and liver tissues were separately collected from 4 female and 4 male one-year-old (1 Y) individuals. The brain tissue of fish at different stages, including three-month-old (3 M), five-month-old (5 M), eight-month-old (8 M), one-year-old (1 Y), 1.5-year-old (1.5 Y), and two-year-old fish (2 Y), were also picked. Furthermore, the embryos at various classic developmental periods such as cleavage period, blastocyst period, gastrula period, segmentation period, pharyngula period and early larval were also harvested. They were stored at −20 °C in RNA preservation solution (TaKaRa, Tokyo, Japan) or 100% ethanol for DNA/RNA extraction, respectively.

### 2.4. DNA Extraction, RNA Extraction, and cDNA Synthesis 

The genomic DNA was extracted using TIANamp Marine Animal DNA Kit (TIANGEN, Beijing, China). Total RNA was extracted by TRIzol (Invitrogen, Waltham, MA, USA) according to standard protocol. The quality, concentration, and integrity of DNA/RNA were determined by agarose gel electrophoresis and NanoVuePlus (GE Healthcare, Little Chalfont, Buckinghamshire, UK). The PrimeScript^TM^ RT reagent kit with gDNA Eraser (Perfect Real Time) (TaKaRa, Tokyo, Japan) was used to make cDNA. CDS of *zbed1* was cloned using primers (*zbed1*-cds-F and *zbed1*-cds-R) (Table 1). The PCR amplification was carried out for 10 s at 98 °C, followed by 40 cycles at 68 °C for 30 s, and 7 s at 72 °C. The PCR product was subcloned to pEASY-T1 vector and sequenced by Qingdao Ruibo Company (Ruibo, Beijing, China).

### 2.5. Identification of Three Multi-Copy Genes of Cs-zbed1 Gene and dPCR

The design of taqman probes and primers targeting the genome sequences of *zbed1* and myosin heavy chain 6 (*myh6*), a single-copy internal reference gene in teleost [38], was conducted by Sangon Biotech Company (Sangon, Shanghai, China). We validated the amplification efficiency of Taqman probes and primers by qPCR. The taqman–PCR reaction system contained 1 μL of DNA, 2.5 μL of each primer, 0.2 μL of taqman probe, 10 μL of KOD mix, and 3.4 μL of ddH_2_O. Next, PCR was performed for 10 s at 98 °C, followed by 40 cycles of 5 s at 58 °C and 30 s at 68 °C.

Furthermore, the digital PCR (dPCR) experiment was performed using the Naica Lite Automatic Microchip Digital PCR System (Stilla Technologies, Paris, France). The amplification system in sapphire chips comprised 25 μL of perfeCTa Multiplex qPCR Toughmix, 2.5 μL of taqman primer, 2.5 μL of Fluorescein, 0.625 μL of taqman probe, 5 μL of DNA, and 11.875 μL of ddH_2_O. After incubating at 95 °C for 3 min, the samples were denatured for 10 s at 95 °C for 40 cycles, and then annealed at 58 °C for 1 min. The number of droplets in sapphire chips were imaged by the Naica Prism2 reader and the fluorescence data were measured by Crystal Miner_v3.1.6.3 software to determine the positivity or negativity. Subsequently, the fluorescence intensity distribution of the droplet and the copy number of each sample were further analyzed by Crystal Reader_Prism3_v3.1.6.3.SP1 software. Finally, the copy number of the target gene was determined by comparing it to the copy number of the internal reference gene.

### 2.6. Phylogenetic Analysis of ZBED Family

To conduct a phylogenetic analysis of the ZBED family, the ZF_BED (PF02892) and Dimer_Tnp_hAT (PF05699) domains of PFAM database were firstly used to search in *C. semilaevis* genome. Combining BLAST and HomoloGene search in the NCBI database, ZBED family members of *C. semilaevis*, and other teleosts and mammals were obtained and submitted to MEGA 7.0 software for phylogenetic tree construction with the neighbor joining method. Furthermore, the tree was modified and visualized by EvolView Old versions Online website (https://www.evolgenius.info/evolview/#/treeview, accessed on 6 December 2023). The protein–protein interaction (PPI) network for ZBED family members based on mammals and fish was built according to the STRING database (https://string-db.org/cgi, accessed on 10 April 2023), with a confidence level of 0.25.

### 2.7. Expression Patterns of zbed1

We designed qPCR primers (*zbed1*-qPCR-F/R) (Table 1) for detecting the expression patterns of *C. semilaevis zbed1*. β-actin was utilized as an internal reference gene. The reaction was conducted using the Ex-Taq with SYBR^®^ Green Realtime PCR Master Mix (TOYOBO, Osaka, Japan) on a 7500 fast real-time PCR system (ABI, Los Angeles, CA, USA). The 2^−ΔΔCt^ method was applied to determine the relative mRNA expression level. Four different female individuals were used as biological repeats and the data were analyzed with SPSS 25.0 (IBM Corp, Armonk, NY, USA) using one-way ANOVA and multiple comparison by the Wohler and Duncan methods, and *p*-value < 0.05 was considered the threshold for statistical significance.

### 2.8. Promoter Activity Analysis of zbed1

The *Zbed1* promoter region was amplified using primers (*zbed1*-pro-F, *zbed1*-pro-R) and cloned into HindIII-digested pGL3-Basic with the TOROIVD^®^ One Step Fusion Cloning MIX SeamLess cloning kit (TOROIVD, Shanghai, China) to generate the recombinant plasmid pGL3-*zbed1*-pro.

HEK 293T cells were separately transfected with the plasmids pGL3-*zbed1*-pro, pGL3-control, and pGL3-basic using Lipo8000^TM^ Transfection Reagent at 800 ng per well in 24-well plates. We used a 40 ng/well concentration of the pRL-TK plasmid as an internal reference. The firefly and Renilla luciferase activities in these cells were measured by using the Dual-Luciferase Reporter Gene Assay Kit (Beyotime, Shanghai, China) and a Varioskan Flash spectral scanning multimode reader (Thermo, Vantaa, Finland) after 48 h. Triplicates of each experiment were performed. The data were analyzed with SPSS 25.0 (IBM Corp, Armonk, NY, USA) using *t*-test. The data of each co-transfection were compared with the original promoter and *p*-value < 0.05 was considered the threshold for statistical significance.

Furthermore, the possible transcription factors binding to *zbed1* promoter were predicted using online tools PROMO (http://alggen.lsi.upc.es/cgi-bin/promo_v3/promo/promoinit.cgi?dirDB=TF_8.3, accessed on 6 October 2022) and JASPAR (http://jaspar.genereg.net/, accessed on 6 October 2022). The CDS of candidate transcription factors were cloned and ligated to pcDNA3.1. The binding site mutations were performed using a rapid site-directed mutagenesis kit (TIANGEN, Beijing, China). The co-transfection of pGL3-*zbed1*-pro and transcription factor plasmids were carried out and the luciferase reporter assay was processed as described above.

### 2.9. Design and Transfection of RNAi in Female C. semilaevis Brain Cell Lines

Based on the *zbed1* mRNA sequences, one siRNA (Table 1) was designed and ordered from Guangzhou RiboBio Co., Ltd. (Ribobio, Guangzhou, China). By using the riboFECT^TM^ CP Transfection Kit (Ribobio, Guangzhou, China), the negative control (RNAi-NC), positive control (RNAi-cy3), and one siRNA for *zbed1* were transfected into female brain cells. We diluted 3 μL of siRNA (20 μM) with 60 μL of CP buffer and 5 μL of CP reagent before adding it to each well of a 12-well plate. A total of 48 h after transfection, total RNA extraction, cDNA synthesis, and qPCR experiment were carried out with the above-described methods. The data were analyzed with SPSS 25.0 (IBM Corp, Armonk, NY, USA) using *t*-test. The data of each downstream gene were compared with NC and *p*-value < 0.05 was considered the threshold for statistical significance.

## 3. Results

### 3.1. Multiple Copies of zbed1 on W Chromosome and Phylogenetic Tree of ZBED Family

By screening domains ZF_BED and Dimer_Tnp_hAT among the *C. semilaevis* genome, three copies of the *zbed1* gene were identified on chromosome W (Figure 1). Briefly, three copies of the *zbed1* gene were located on the W chromosome of *C. semilaevis*: Gene ID 103397026 at W_5463004-5464806; Gene ID 107990198 at W_7230771-7232573; and Gene ID 103397195 at W_11375464-11377266. PCR cloning and sequencing confirmed that the length for each *zbed1* gene was 1802 bp, with three exons and two introns. In addition, the cDNA for each *zbed1* gene was 1491 bp, encoding 496 amino acids.

To verify whether they were three copies of *zbed1* in the *C. semilaevis* and whether there were gametologs on the chromosome Z, taqman primers and probes were designed based on *zbed1* and *myh6*, a single-copy gene located on the chromosome 7 of *C. semilaevis*. Firstly, PCR experiment by taqman primers revealed that only female-specific target bands were amplified for *zbed1*, while for the autosome gene myh6, female and male target bands were both observed from the genomic DNA template (Figure 2A). Furthermore, digital PCR was employed for the confirmation of the gene copy number of *zbed1* in the female individuals. The obvious distinction was detected between the number of positive droplets and negative droplets (no template control, NTC) generated by the digital PCR system in the same detection channel, demonstrating that the system exhibited consistent droplet generation and reliable repeatability (Figure 2B). The results showed that the copy number of the *zbed1* gene was three times or more that of *myh6* in three female individuals (Figure 2C).

As expected, the phylogenetic tree of ZBED homologues from different species displayed eleven sub-branches (Figure 3A). Then, the sub-branches of Ac (ZBED1, ZBED4, ZBED6cl, ZBED6, ZBEDX, ZBED2, and ZBED3) and Buster (ZBED7, ZBED8, ZBED5, and ZBED9) transposons were clustered into two big branches. In addition to ZBED1, ZBED4 and ZBEDX, which were widely found in fish, we also retrieved several fish ZBED5-like sequences from the NCBI database, and the phylogenetic results revealed that these sequences gathered with mammalian ZBED9 proteins, which suggested that they might be fish ZBED9. Similarly, three ZBED4-like of *C. semilaevis* were grouped into the ZBEDX sub-branch, indicating that they may be correctly called ZBEDX. Thus, only four ZBED1, one ZBED4, and four ZBEDX were identified in *C. semilaevis*, of which the three copies of ZBED1 were located on chromosome W and ZBED1-like (ZBED1l) was on chromosome 10 (Figure 3A, Table 2). In each sub-branch of ZBED1, ZBED4, and ZBEDX, the sequences of *C. semilaevis* and other fishes clustered together, followed by clustering with mammalian proteins.

The expression heatmap (Figure 3B) based on our previous two-year-old transcriptomic data [23] revealed that *zbed1* was predominantly expressed in the female brain and pituitary. *Zbed1l* was highly expressed in the pituitary gland and male gonad. The expression of *zbed4* mainly focused on the male pituitary gland, brain, and gonad. As for *zbedx*, a high expression level was detected in the gonad and liver.

### 3.2. Interaction Network Analysis of ZBEDs

For a better understanding of ZBED’s biological activity and complex regulatory network, a protein–protein interaction (PPI) network was constructed (Figure 4). This network predicted the interactions among 11 members of the ZBED family (ZBED1-9, ZBEDX, and ZBED6CL) and 25 other proteins, including IGF2 and its associated members (IGF1R, IGF2R, IGFBP1-3, IGFBP5-6), several zinc finger proteins including zinc finger protein 396 (ZNF396) and ZNF444, dehydrogenase/reductase X-linked (DHRSX), Karyopherin subunit alpha 1 (KPNA1), Small ubiquitin-related modifier 3-like (SUMO3A), and Acetylserotonin O-methyltransferase-like (ASMTL). Importantly, ZBED1 exhibited a complex interaction with other ZBED members. However, ZBEDX did not interact with other family members.

### 3.3. The Expression of zbed1 in Different Tissues and at Different Developmental Stages

The qPCR results indicated that *zbed1* was highly expressed at the blastocyst and gastrula periods of embryonic development (Figure 5A). Furthermore, the *zbed1* gene exhibited widespread expression in different tissues of female fish, with particularly high levels in the pituitary and brain (Figure 5B). The expression pattern of *zbed1* in the female brain at different developmental stages showed that the highest level was detected at 2 Y (Figure 5C).

### 3.4. Knock-Down Effects on zbed1 and Other Related Genes by RNAi Transfection in Brain Cells

Female *C. semilaevis* brain cells were used for RNAi experiments to investigate the potential knock-down impact of *zbed1*. Within the exon region, one siRNA was designed. A high transfection efficiency (>90%) of RNAi-cy3 was observed (Figure 6A). qPCR analysis was performed to evaluate the expression levels of *zbed1* and its related genes, such as piwi-like RNA-mediated gene silencing 1 (*piwil1*), estrogen receptor 2 (*esr2*), wingless-type MMTV integration site family, member 7B (*wnt7b*), TATA box binding protein (*tbp*), cyclin-dependent kinase 2 (*cdk2*), cyclin-dependent kinase 4 (*cdk4*), cyclin-dependent kinase 6 (*cdk6*), cyclin G1 (*ccng1*), and cyclin Dx (*ccndx*). In female brain cells, siRNA1 exhibited a significant knock-down effect (Figure 6B). Following this, *piwil1*, *esr2* and *wnt7b* demonstrated up-regulation, while *tbp*, *cdk2*, *cdk4*, *cdk6*, *ccng1* and *ccndx* displayed down-regulation (Figure 6C).

### 3.5. Site-Directed Mutagenesis of zbed1 Promoter Activity and Transcription Factor Sites

A promoter region of *zbed1* was cloned at the upstream 11–2117 bp of the start codon and named *zbed1*-pro. In PROMO and JASPAR analysis, several transcription-factor-binding sites were predicted, including Yin Yang 1A (*yy1a*), signal transducer and activator of transcription 5A (*stat5a*), CCAAT/enhancer-binding protein Alpha (*c/ebpα*), SRY-related high-mobility-group box protein 2 (*sox2*), and POU domain, class 1, transcription factor 1 (*pou1f1*), activator protein 1 (AP-1) family members, include two major types of protein transcription factors (*junb* and *fos*), as well as *myogenin* in the MyoD family (Figure 7A).

The dual-luciferase assay demonstrated that the *zbed1* promoter was highly active in promoting firefly luciferase transcription (Figure 7B). The subsequent co-transfection assay revealed that the *zbed1* promoter activity exhibited significant up-regulation upon binding with the transcription factors *yy1a* and *c/ebpα*, whereas a significant decrease in promoter activity was observed upon binding with *sox2*, *pou1f1a*, *myogenin*, and *junb*. 

Given the crucial involvement of *c/ebpα* in growth and development, an additional investigation was conducted to ascertain the influence of *c/ebpα* transcription factor on the *zbed1* promoter. It involved the introduction of mutation for the two *c/ebpα* sites on the promoter, followed by co-transfection with *c/ebpα* plasmid. The results revealed that no up-regulation effect was observed once the two binding sites were mutated (Figure 7C).

## 4. Discussion

More than 600 fish species exhibit sexual size dimorphism (SSD) [22]. This phenomenon often leads to serious male or female growth disadvantages, resulting in high breeding costs and limiting the feasibility of the fish-breeding industry. To investigate the related mechanism mediating SSD will be meaningful for the molecular breeding of fish with high-quality growth traits.

The identification of *zbed1* is particularly interesting because it is a homologue in *Drosophila* (DREF) and human (hDREF/*zbed1*) [39]. Through our research, we found that *zbed1* had three identical copies on the W chromosome of *C. semilaevis*, and no homologous gene was found on the Z chromosome. There are two important domains in *zbed1*, namely the amino terminal BED zinc finger domain and the carboxyl terminal hATC domain, which provides important structural support for the *zbed1* molecule to distribute in the nucleus with a granular structure [40].

Unlike the eleven ZBED members identified in the mammals, only three members (ZBED1, ZBED4 and ZBEDX) have been identified in the vast majority of fish. The present study revealed the existence a new ZBED member, ZBED9, in this fish species, which broadened the current knowledge regarding the fish ZBED family [33]. More numbers or more functions for duplicate ZBED members would be found with increasing research on fish genomics.

Consistent with our previous transcriptomic data [24], the qPCR results revealed that *zbed1* was significantly expressed in the female brain and pituitary. Differently, human *zbed1* was widely expressed in all tissues from three primary germ layers, with a high expression in the digestive and reproductive systems [41]. Therefore, we speculated that the regulation of the W chromosome gene *zbed1* on female-biased SSD in *C. semilaevis* may be realized by its high specific expression in female brain neurons and pituitary hormone cells, similar to the sexual dimorphic singing behavior of birds [14] and female-biased SSD of *drosphila* [16].

It was also found that *C. semilaevis zbed1* was significantly highly expressed in the blastocyst period and gastrula period and reached its peak in the gastrula period. In zebrafish, the early development of the neural tube begins from the blastocyst period [42,43], and the formation of neuroectoderm and neural plate in the gastrula period lays the foundation for the neural tube [44,45]. Thus, it was suggested that *zbed1* was involved in the early development of the nervous system of *C. semilaevis*.

The subsequent analysis of transcriptional regulation suggested that *c/ebpα* may serve as a crucial transcription factor in activating the transcriptional activity of *zbed1*. As a conserved transcription factor involved in cellular growth and differentiation [46], *c/ebpα* plays a role in the early differentiation of gonads and acts as a suppressor of the male determinant gene *dmrt1* in *C. semilaevis* [47]. This implies that *c/ebpα* has an opposing effect on *dmrt1* and *zbed1*, leading us to propose that *zbed1* may be a significant gene associated with the regulation of female behavior.

As for the knockdown effect of *zbed1* on cell proliferation and growth performance, important genes derived from the cell cycle, Wnt pathway and other pathways according to our previous DAP-seq analysis [24] were determined to be regulated by *zbed1* RNAi in the brain cell line. *Wnt7b*, one important factor of the Wnt pathway, was upregulated after the knockdown of *zbed1*. A Wnt transduction abnormality will cause nervous system developmental defects and lead to many neurological diseases [48,49,50,51]. It is suggested that *zbed1* may regulate the development of the nervous system of *C. semilaevis* by regulating the expression of *wnt7b*. In addition, as the main receptor for estrogen, the expression of *esr2* was also increased after the knockdown of *zbed1* in the brain cells. In mammals, *esr2* may inhibit granulosa cell proliferation by mediating the effect on *cyclin-dependent kinase inhibitor 1a* (*cnkn1a*) [52]. It was hypothesized that *zbed1* may exert control over cell proliferation by reducing the expression of *esr2*.

Notably, the knockdown of *zbed1* resulted in the decrease in several genes involved in the cell cycle, including *cdk2*, *cdk4*, *cdk6*, *ccng1*, and *ccndx*. The activation of the cell cycle pathway has been proposed as a significant factor contributing to the SSD in *C. semilaevis* [53]. Our findings suggest that *zbed1*, an upstream transcription factor located on the W chromosome, may regulate the activation of the cell cycle.

## 5. Conclusions

In this study, we identified three copies of *zbed1* on the W chromosome of *C. semilaevis*, which displayed a female-biased expression pattern in the brain and pituitary. The experiments demonstrated that the transcription factor *c/ebpα* can activate the transcriptional activity of the *zbed1* promoter. It is also suggested that *zbed1* may play a vital function in cell proliferation by regulating *esr2*, *piwil1*, the cell cycle, and the Wnt pathway. Further investigations into the interaction among *esr2*, the Wnt pathway, and *zbed1* will contribute to a better understanding of the mechanism underlying sexual size dimorphism in *C. semilaevis*.

## Figures and Tables

**Figure 1 biology-13-00141-f001:**
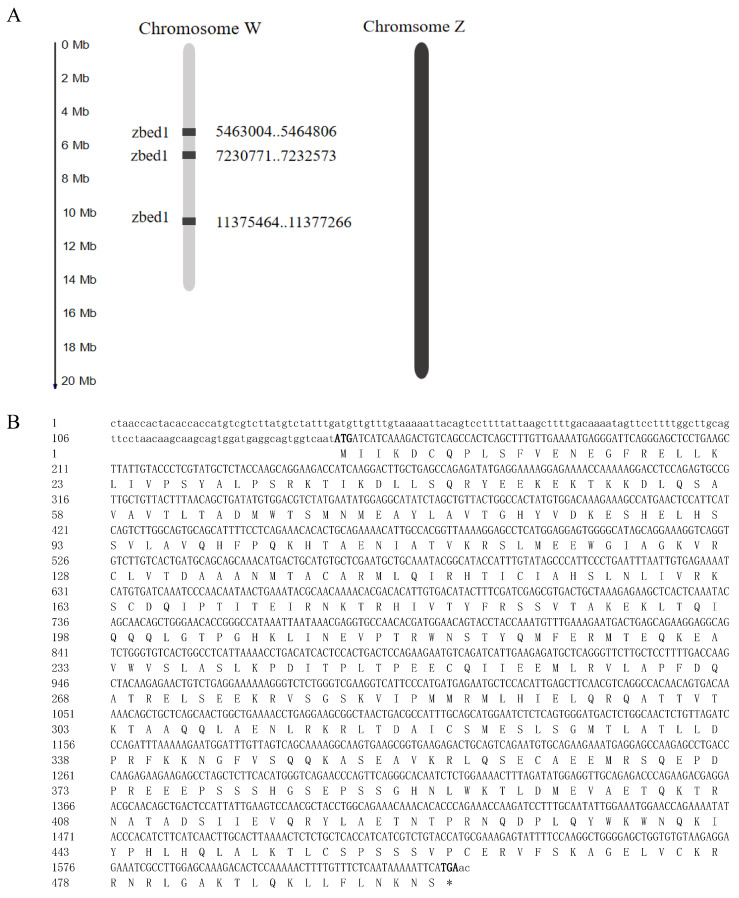
The genomic location and sequences of *Cynoglossus semilaevis zbed1* gene. (**A**) Location of *zbed1* gene on the sex chromosome of *C. semilaevis*. There were three copies of *zbed1* at different locations on the W chromosome and none on the Z chromosome. (**B**) Sequence information for *C. semilaevis zbed1* cDNA and protein. The start and stop codons are shown in bold.

**Figure 2 biology-13-00141-f002:**
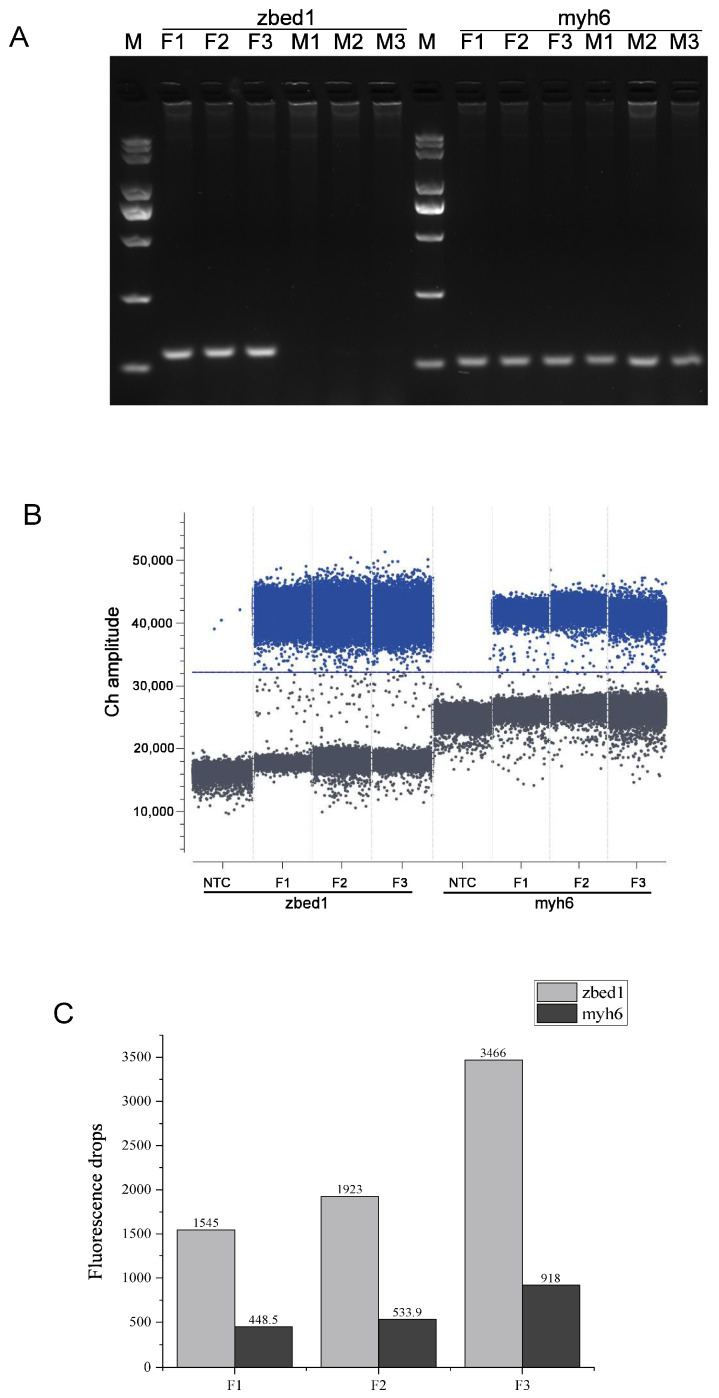
The specificity of *zbed1* in the females and the confirmation of its copy number. (**A**) PCR experiment result for the detection of *zbed1* and *myh6* from genomic DNA. F1-3 and M1-3 indicated three female and male individuals, respectively. M was the maker. (**B**) Droplet distribution map. *Zbed1*, *myh6* and NTC droplets were clearly distinguished under the detection channel. The blue line showed the fluorescence threshold. (**C**) The copy numbers of *zbed1* and the single-copy gene *myh6* in three female individuals.

**Figure 3 biology-13-00141-f003:**
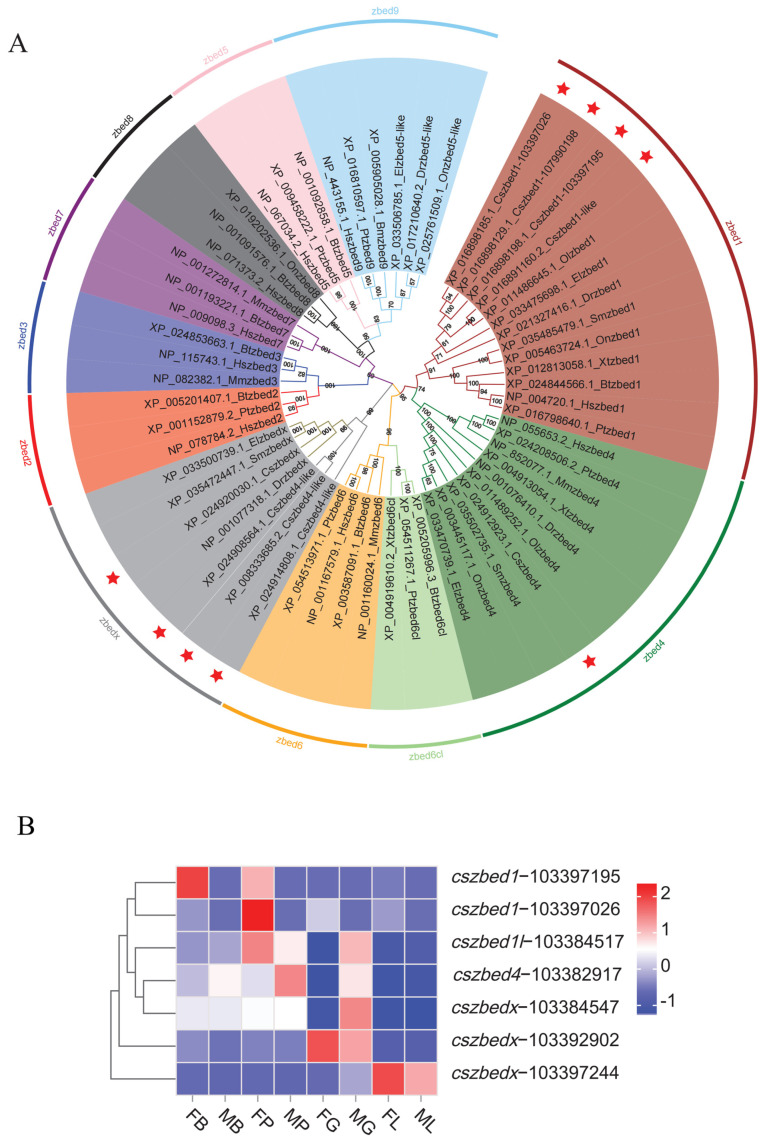
Phylogenetic tree and expression heatmap of ZBED family genes. (**A**) Phylogenetic tree of ZBEDs from *Cynoglossus semilaevis* (Cs), *Homo sapiens* (Hs), *Mus musculus* (Mm), *Danio rerio* (Dr), *Oryzias latipes* (Ol), *Oreochromis niloticus* (On), *Epinephelus lanceolatus* (El), *Scophthalmus maximus* (Sm), *Xenopus tropicalis* (Xt), *Bos taurus* (Bt), and *Pan troglodytes* (Pt). The percentage of replicate trees in which associated taxa clustered together in the bootstrap test (1000 replicates) is shown next to the branches. Eleven sub-clusters were indicated in different colors. Then, the sub-branches of Ac (ZBED1, ZBED4, ZBED6cl, ZBED6, ZBEDX, ZBED2, and ZBED3) and Buster (ZBED7, ZBED8, ZBED5, and ZBED9) transposons were clustered into two big branches. In each sub-branch of ZBED1, ZBED4, and ZBEDX, the sequences of C. semilaevis and other fishes clustered together, followed by clustering with mammalian proteins. CsZBEDs were marked in red stars. (**B**) Heatmap of *zbed* mRNA abundances in different tissues of *C. semilaevis* female and male individuals. FB: female brain, MB: male brain, FP: female pituitary, MP: male pituitary, FG: female gonad, MG: male gonad, FL: female liver, ML: male liver.

**Figure 4 biology-13-00141-f004:**
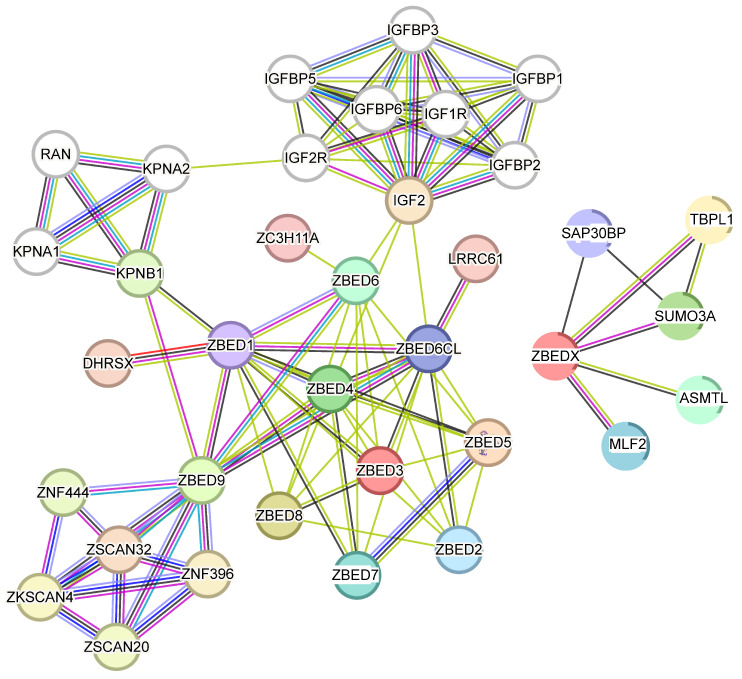
Protein interaction network among ZBED’s family members. Nodes indicate the interactive proteins; edges indicate both functional and physical protein associations; and different colors indicate the various types of interaction evidence.

**Figure 5 biology-13-00141-f005:**
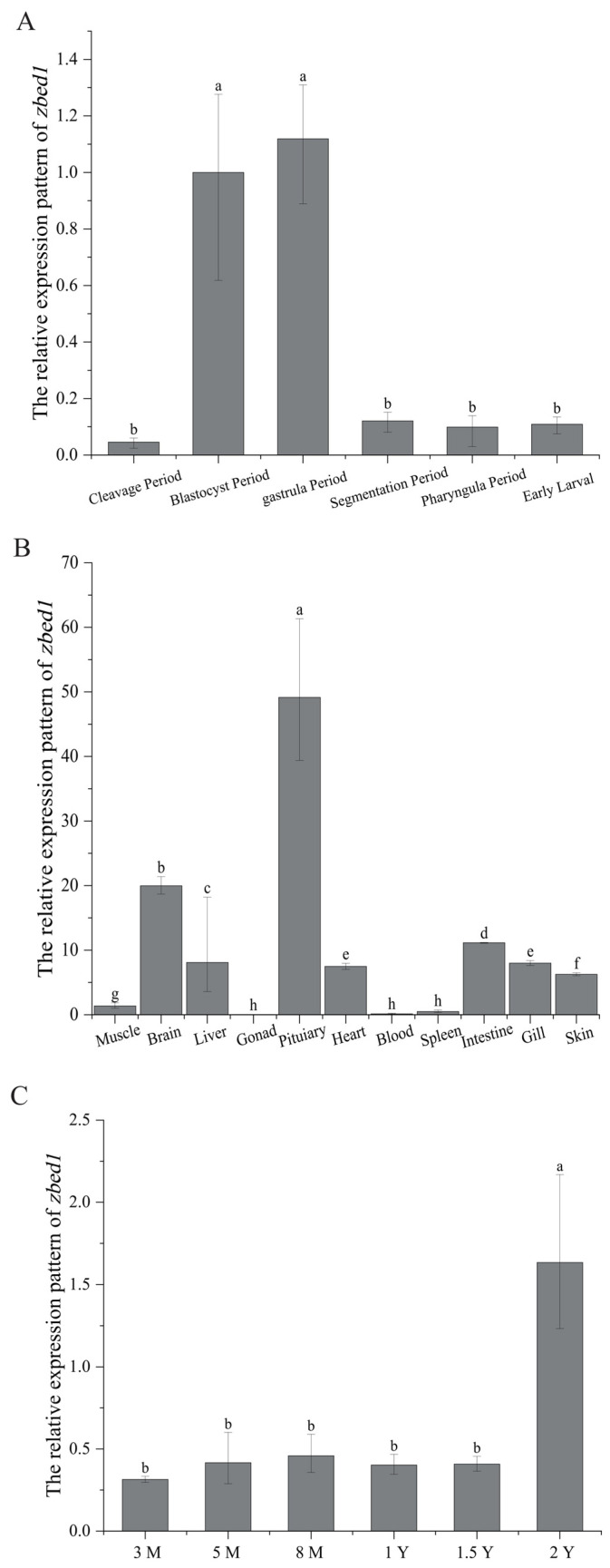
Spatiotemporal expression patterns of *zbed1* gene in *C. semilaevis*. The letters a–h represent significance. There are significant differences between columns with different letters in each picture. (**A**) Relative mRNA expression patterns of *zbed1* in *C. semilaevis* embryos at five periods (cleavage, blastocyst, gastrula, segmentation, pharyngula and early larval). (**B**) Relative mRNA expression patterns of *zbed1* in 11 tissues (brain, pituitary, gill, gonad, heart, intestine, kidney, liver, muscle, skin, and spleen) of female *C. semilaevis*. (**C**) Relative mRNA expression patterns of *zbed1* in the brain of female *C. semilaevis* at six developmental stages including three-month-old (3 M), five-month-old (5 M), eight-month-old (8 M), one-year-old (1 Y), 1.5-year-old (1.5 Y), and two-year-old fish (2 Y). The data were analyzed with SPSS 25.0 (IBM Corp, Armonk, NY, USA) using one-way ANOVA and multiple comparison by Wohler and Duncan methods, and *p*-value < 0.05 was considered the threshold for statistical significance.

**Figure 6 biology-13-00141-f006:**
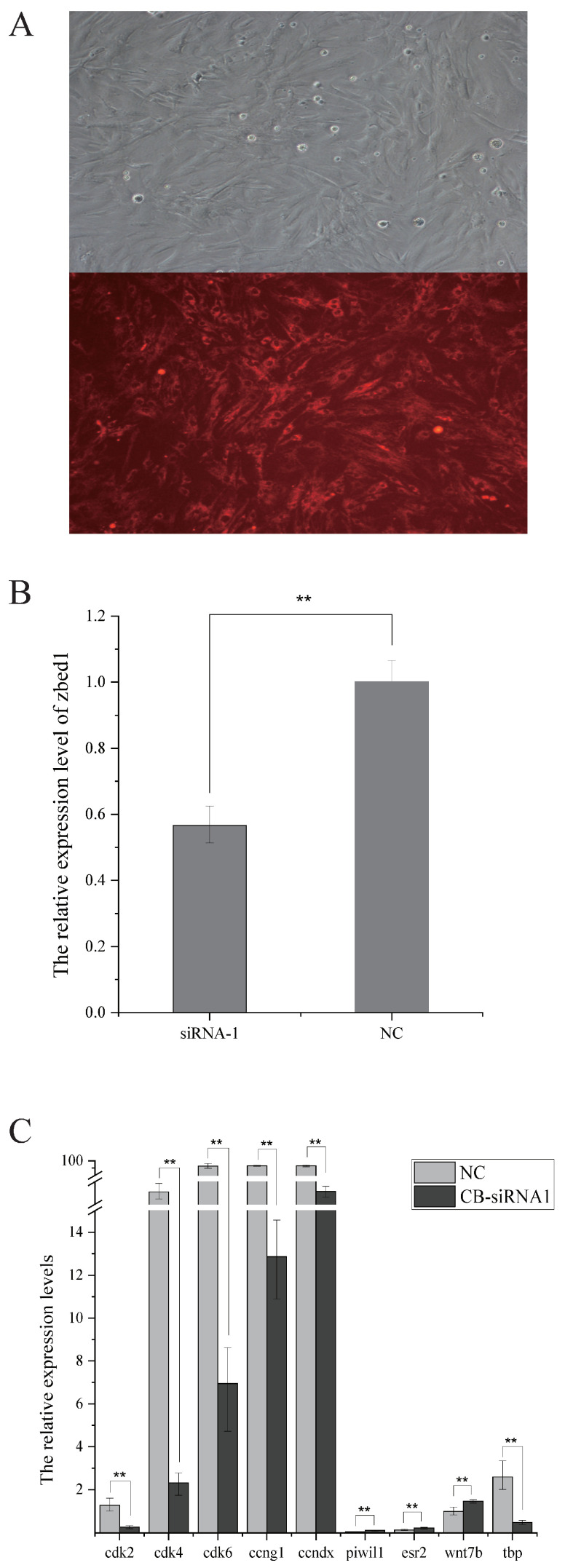
The knockdown effect of *zbed1* on the female *C. semilaevis* brain cells. (**A**) RNAi transfection efficiency in brain cells. (**B**) Interference efficiency of *zbed1* siRNA. (**C**) The expression patterns of genes in female brain cells after transfection with *zbed1* siRNA. The data in (**B**,**C**) were analyzed with SPSS 25.0 (IBM Corp, Armonk, NY, USA) using *t*-test. The data of each downstream gene were compared with NC and *p*-value < 0.05 was considered the threshold for statistical significance (double asterisk, *p* < 0.01).

**Figure 7 biology-13-00141-f007:**
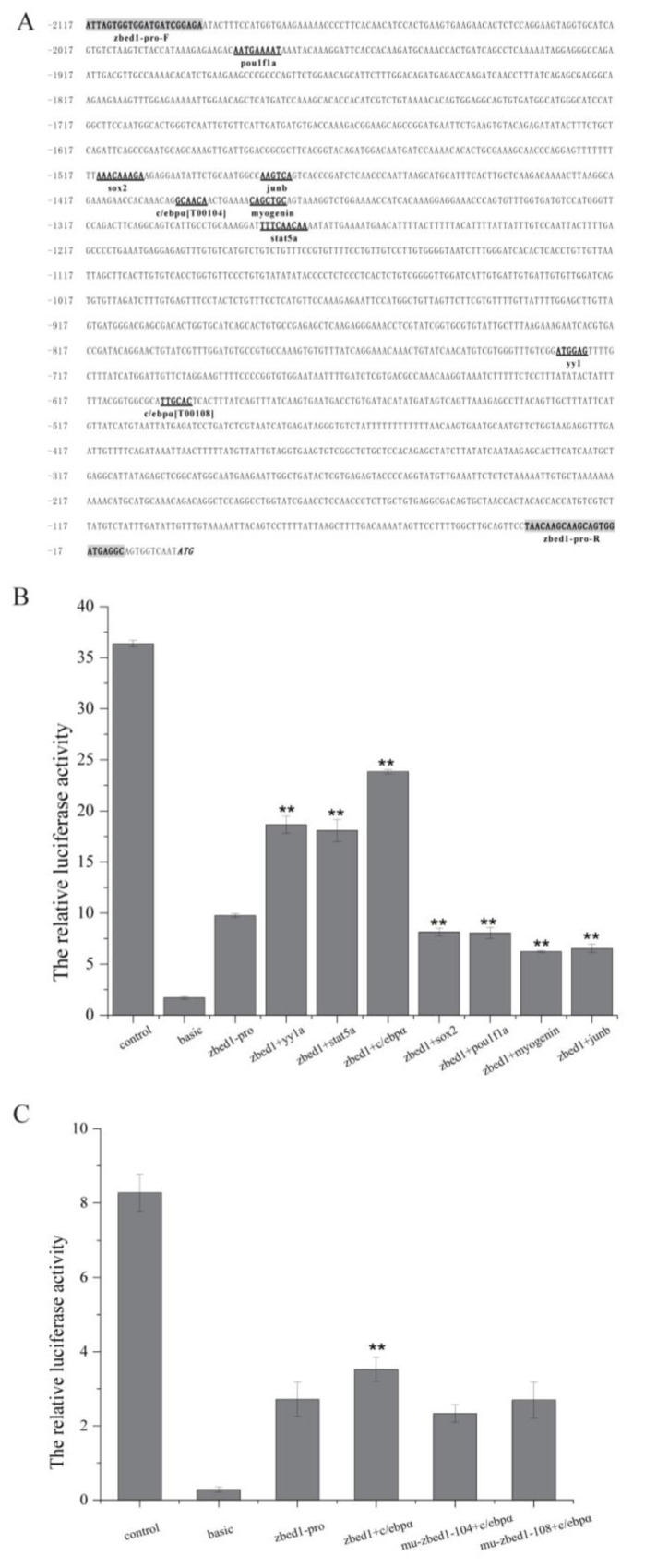
Structure, activity, and transcription factor analysis of *C. semilaevis zbed1* promoter. (**A**) Sequence structure and putative transcription-factor-binding sites (i.e., *pou1f1a*, *sox2*, *junB*, *c/ebpα*, *myogenin*, *yy1*, and *stat5a*) on *zbed1* promoter. Two primers of the promoter are shown in shadow. (**B**) Transcription activity of *zbed1* promoter and co-transfection with transcription factors in HEK 293T cells. (**C**) The luciferase activity after co-transfection with the transcription factor *c/ebpα* and mutated *zbed1* promoter. The significance is indicated by asterisks. The data in (**B**,**C**) were analyzed with SPSS 25.0 (IBM Corp, Armonk, NY, USA) using *t*-test. The data of each co-transfection were compared with the original promoter and *p*-value < 0.05 was considered the threshold for statistical significance (double asterisk, *p* < 0.01).

**Table 1 biology-13-00141-t001:** The primers used in present study.

Primers	Information	Sequences (5′–3′)
Cs-SEX-F	sex detection	CCTAAATGATGGATGTAGATTCTGTC
Cs-SEX-R	sex detection	GATCCAGAGAAAATAAACCCAGG
*zbed1*-cds-F	CDS cloning	ATGATCATCAAAGACTGTCAGCC
*zbed1*-cds-R	CDS cloning	TCATGAATTTTTATTGAGAAACA
*zbed1*-pro-F	promoter cloning	AGATCTGCGATCTAAGTAAGCTATTAGTGGTGGATGATCGGAGA
*zbed1*-pro-R	promoter cloning	CAACAGTACCGGAATGCCAAGCTGCCTCATCCACTGCTTGCTTGTTA
*zbed1*-mu-*c/ebpα*104-F	promoter mutation	AAAGAACCACAAACAGCGATCAACT
*zbed1*-mu-*c/ebpα*104-R	promoter mutation	AGTTGATCGCTGTTTGTGGTTCTTT
*zbed1*-mu-*c/ebpα*108-F	promoter mutation	TGGCGCATACCAGTCACTTTATCAG
*zbed1*-mu-*c/ebpα*108-R	promoter mutation	CTGATAAAGTGACTGGTATGCGCCA
*zbed1*- site1	RNAi site1	GCAACAGCTGACTCCATTA
RNAi-NC	negative control (nc)	CTGAAGATCCGGCTCATCA
*zbed1*-qPCR-F	qPCR	CTCCAGAGTGCCGTTGC
*zbed1*-qPCR-R	qPCR	GTTCATGGCTTTCTTTGTCC
*actin*-F	qPCR	TTCCAGCCTTCCTTCCTT
*actin*-R	qPCR	TACCTCCAGACAGCACAG
*esr2*-qPCR-F	qPCR	GATTAGGAGAAGGTGGAGAAGG
*esr2*-qPCR-R	qPCR	GGTAACCAGAGGCATAGTCGTG
*ccng1*-qPCR-F	qPCR	AGTGACTACGCCAACACCAAAT
*ccng1*-qPCR-R	qPCR	GATGGTAGGCAGATGAGCGATT
*ccndx*-qPCR-F	qPCR	CCTTGTCCTTGCCTATCTC
*ccndx*-qPCR-R	qPCR	GACGCCTCAAAGTTGTTCT
*piwill*-qPCR-F	qPCR	CATCCAACTGTCGGCCAACTAT
*piwill*-qPCR-R	qPCR	TCGGCAATCTATTAGGCAGGAA
*cdk4*-qPCR-F	qPCR	CGCCAGTATGCAGTATGA
*cdk4*-qPCR-R	qPCR	TCTTGAGCAGAGCCACCT
*cdk2*-qPCR-F	qPCR	CACTGGTATCCCTCTGCC
*cdk2*-qPCR-R	qPCR	GAAGTCGGCGAGTTTGAT
*cdk6*-qPCR-F	qPCR	TACCACCCGAGACCATTA
*cdk6*-qPCR-R	qPCR	TAGATTCGAGCCAGACCA
*tbp*-qPCR-F	qPCR	AAACAGTAACAGGCTCCAC
*tbp*-qPCR-R	qPCR	TCCAGTTTACAGCCAAGAT
*wnt7b*-qPCR-F	qPCR	AGCAGCATTCACCTACGC
*wnt7b*-qPCR-R	qPCR	CTTCCAGCCTTCCTCTTG
*zbed1*-taqman-F	Taqman primer	CTCTGGCAACTCTGTTAGATCC
*zbed1*-taqman-R	Taqman primer	GCTCTTGGCTCCTCATTTCT
*zbed1*-taqman-probe	Taqman probe	AAAGGCAAGTGAAGCGGTGAAGAGAC 5′6-FAM, 3′BHQ1
*myh6*-taqman-F	Taqman primer	ACAAGTGGCTTCCTGTCTATG
*myh6*-taqman-R	Taqman primer	GCGTTATCGGAGATGGAGAAA
*myh6*-taqman-probe	Taqman probe	TAAGAAGAGAAGCGAGGCTCCACCTC 5′6-FAM, 3′BHQ1

**Table 2 biology-13-00141-t002:** The ZBED family members identified from *C. semilaevis* genome.

Name	Gene ID	Protein ID	Gene Length (bp)	ORF Length (bp)	Amino Acid Length (aa)	Chr	Location	No. of Exons	No. of Introns
*zbed1*	103397026	XP_016898185.1	1802	1491	496	W	5,463,004–5,464,806	3	2
*zbed1*	103397195	XP_016898198.1	1802	1491	496	W	11,375,464–11,377,266	3	2
*zbed1*	107990198	XP_016898129.1	1802	1491	496	W	7,230,771–7,232,573	3	2
*zbed1l*	103384517	XP_016891160.2	1990	1537	511	10	2,017,183–2,0191,72	2	1
*zbed4*	103382917	XP_024912923.1	7442	3750	1249	8	26,160,975–26,168,416	3	2
*zbedx*	103384547	XP_024914808.1	7167	2163	720	10	2,351,855–2,359,021	6	5
*zbedx*	103397244	XP_008333685.2	52448	1161	386	W	13,905,364–13,957,811	2	1
*zbedx*	112486373	XP_024908564.1	1318	1089	362	W	10,396,182–10,397,499	2	1
*zbedx*	103392902	XP_024920030.1	69214	1550	499	17	6,577,362–6,646,575	7	6

## Data Availability

Data are contained within the article.

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
