# Peer review of "Three Copies of zbed1 Specific in Chromosome W Are Essential for Female-Biased Sexual Size Dimorphism in Cynoglossus semilaevis"

_biology, 2024, doi:10.3390/biology13030141_

Round 1

Reviewer 1 Report

Comments and Suggestions for Authors

The paper aimed to explore the potential mechanism involved in Chinese tongue sole sexual size dimorphism (SSD) by focusing on zbed1, a transcription factor specific located in the chromosome W. Given that the mechanism for fish SSD is still mysterious, the present study is meaningful and helpful for further understanding fish SSD. In my opinion, the paper could be considered for publication after the following minor revisions:

1. “Recently” in Line 67 may not be appropriate due to that the reference of Rice et al. was published in 1984.   

2. The sentence in Line 78-79 is confused and needed to revise.

3. In the paragraph of Line 91-99, more information about the function of zbed1 in other species is suggested to provide.

4. The sentence in Line 115-117 is repeated with previous content and needs to be deleted.   

5. “a small three pairs of siRNAs” in Line 211 seems contradictory to “one siRNA for zbed1” in Line 214.

6. Given that the funding information has been provided, the acknowledgments part is suggested to be simplified or deleted.

7. The references style including letter case should be adjusted into a uniform format. 

Comments on the Quality of English Language

Minor editing of English language required

Author Response

Dear Editors and Reviewers:

Thank you for giving us the opportunity to submit a revised draft of the manuscript for publication in Biology. We appreciate the time and effort that you dedicated to providing feedback on our manuscript and are grateful for the insightful comments on and valuable improvements to our paper. We have incorporated the suggestions made by the reviewers. Those changes are highlighted in the manuscript. Please see below for a point-by-point response to the reviewers’ comments and concerns.

Question: “Recently” in Line 67 may not be appropriate due to that the reference of Rice et al. was published in 1984.

Response: We sincerely appreciate the valuable comments. We changed the words and corrected the inappropriate words in the manuscript to eliminate misunderstandings in the revised manuscript.

Question: The sentence in Line 78-79 is confused and needed to revise.

Response: Your suggestion really means a lot to us. The description about female-biased SSD regulation in drosophila has been revised as “In drosphila, sex-determining gene sxl in the chromosome X has also been proven to regulate female-biased SSD by cell- autonomy and non-cell-autonomy mechanisms”.

Question: In the paragraph of Line 91-99, more information about the function of zbed1 in other species is suggested to provide.

Response: As suggested by the reviewer, more information about the function of zbed1 in human has been supplemented.

Question: The sentence in Line 115-117 is repeated with previous content and needs to be deleted.

Response: As suggested by the reviewer, we deleted the repetitive sentences here.

Question: “a small three pairs of siRNAs” in Line 211 seems contradictory to “one siRNA for zbed1” in Line 214.

Response: As suggested by the reviewer, we corrected the misrepresentation here and changed “a small three pairs of siRNAs” to “one siRNA”.

Question: Given that the funding information has been provided, the acknowledgments part is suggested to be simplified or deleted.

Response: As suggested by the reviewer, the acknowledgments part have been removed in the revised manuscript.

Question: The references style including letter case should be adjusted into a uniform format.

Response: We sincerely appreciate the valuable comments. We carefully checked the references style and made modifications in the manuscript.

Finally, thanks for your consideration of our manuscript. We would be appreciated if the revised manuscript could be published in Biology.

With best regards,

Na Wang, Ph.D., Professor

Yellow Sea Fisheries Research Institute, Chinese Academy of Fishery Sciences

Qingdao 266071, China

Reviewer 2 Report

Comments and Suggestions for Authors

In general the images should be improved, it is not possible to see them in the sent document.

I suggest expanding the description of the statistical analyses. I consider that the results are not widely described, especially the results of ANOVA.

Comments on the Quality of English Language

Author Response

Dear Editors and Reviewers:

Thank you for giving us the opportunity to submit a revised draft of the manuscript for publication in Biology. We appreciate the time and effort that you dedicated to providing feedback on our manuscript and are grateful for the insightful comments on and valuable improvements to our paper. We have incorporated the suggestions made by the reviewers. Those changes are highlighted in the manuscript. Please see below for a point-by-point response to the reviewers’ comments and concerns.

Question: In general the images should be improved, it is not possible to see them in the sent document.

Response: We sincerely appreciate the valuable comments. We have improved the picture in the manuscript to ensure clarity.

Question: Reviewer suggest expanding the description of the statistical analyses. I consider that the results are not widely described, especially the results of ANOVA. Zbed1's tissue quantitative data does not describe how many samples were taken to perform.

Response: We sincerely appreciate the valuable comments. We have supplemented the statistical description and the results of ANOVA.

Question: Figure 1B should be improved.

Response: We sincerely appreciate the valuable comments. We have improved the text size and clarity of fig1B to ensure the clarity of vector drawings.

Question: Figure 3A is more difficult to understand.

Response: We sincerely appreciate the valuable comments. We have supplemented the description of figure 3A so that readers can better understand it.

Question: Figure 3B's caption does not describe the acronyms for tissues

Response: We sincerely appreciate the valuable comments. We have added the full meaning of the tissue abbreviation mentioned in the picture to the caption of the manuscript.

Finally, thanks for your consideration of our manuscript. We would be appreciated if the revised manuscript could be published in Biology.

With best regards,

Na Wang, Ph.D., Professor

Yellow Sea Fisheries Research Institute, Chinese Academy of Fishery Sciences

Qingdao 266071, China

Reviewer 3 Report

Comments and Suggestions for Authors

The  manuscript Three copies of zbed1 specific in the chromosome W are essential for female-biased sexual size dimorphism in Cynoglossus semilaevis by Yuqi Sun , Xihong Li  , Jiaqi Mai , Wenteng Xu , Jiacheng Wang , Qi Zhang , Na Wang , is well-stuctured and orginised paper.

I read it with an interest. Metohds are adequate and results are clearly represented, gel images are of good quality. The Figures are well structured. The problem of sex determination and genetic control of body size is actual for multiple species across Animal kingdom.   I definitely recommend it for publication. However, I do have minor comments:

- please add a couple of sentences on invertebrates SSD genes diversity

- please compare your results with other Chordates and Invertebrates.

- please look throghout thee text and fix typos.

Author Response

Dear Editors and Reviewers:

Thank you for giving us the opportunity to submit a revised draft of the manuscript for publication in Biology. We appreciate the time and effort that you dedicated to providing feedback on our manuscript and are grateful for the insightful comments on and valuable improvements to our paper. We have incorporated the suggestions made by the reviewers. Those changes are highlighted in the manuscript. Please see below for a point-by-point response to the reviewers’ comments and concerns.

Question: Please add a couple of sentences on invertebrates SSD genes diversity.

Response: We sincerely appreciate the valuable comments. For SSD invertebrates such as spiders, most research focused on the evolution analysis and rare molecular mechanism was available. For drosphila, another invertebrate, except for the classical insulin/IGF pathway (IIS), sex determination gene-sxl has also been proven to regulate female-biased SSD. This information has been supplemented in the introduction.

Question: Please compare your results with other Chordates and Invertebrates.

Response: Your suggestion really means a lot to us. We mentioned the relationships between mammals, birds, amphibians, and fish in the ZBED evolutionary tree. Given that zbed1 gene is mainly described in the humans, we just compared the qPCR results in C. semilaevis with human.

Question: please look throghout thee text and fix typos.

Response: Thanks for your careful checks. In our revised manuscript, the errors have been revised.

Finally, thanks for your consideration of our manuscript. We would be appreciated if the revised manuscript could be published in Biology.

With best regards,

Na Wang, Ph.D., Professor

Yellow Sea Fisheries Research Institute, Chinese Academy of Fishery Sciences

Qingdao 266071, China
